# New Sensor to Measure the Microencapsulated Active Compounds Released in an Aqueous Liquid Media Based in Dielectric Properties in Radiofrequency Range

**DOI:** 10.3390/s21175781

**Published:** 2021-08-27

**Authors:** Juan Angel Tomas-Egea, Pedro J. Fito, Ricardo J. Colom, Marta Castro-Giraldez

**Affiliations:** 1Instituto Universitario de Ingeniería de Alimentos para el Desarrollo, Universitat Politècnica de València, Camino de Vera s/n, 46022 Valencia, Spain; juatomeg@upv.es (J.A.T.-E.); marcasgi@upv.es (M.C.-G.); 2Instituto de Instrumentación para Imagen Molecular, Universitat Politècnica de València, Camino de Vera s/n, 46022 Valencia, Spain; rcolom@eln.upv.es

**Keywords:** sensor, impedance, microencapsulation, radio frequency

## Abstract

In recent years, the general and scientific interest in nutrition, digestion, and what role they play in our body has increased, and there is still much work to be carried out in the field of developing sensors and techniques that are capable of identifying and quantifying the chemical species involved in these processes. Iron deficiency is the most common and widespread nutritional disorder that mainly affects the health of children and women. Iron from the diet may be available as heme or organic iron, or as non-heme or inorganic iron. The absorption of non-heme iron requires its solubilization and reduction in the ferric state to ferrous that begins in the gastric acid environment, because iron in the ferric state is very poorly absorbable. There are chemical species with reducing capacity (antioxidants) that also have the ability to reduce iron, such as ascorbic acid. This paper aims to develop a sensor for measuring the release of encapsulated active compounds, in different media, based on dielectric properties measurement in the radio frequency range. An impedance sensor able to measure the release of microencapsulated active compounds was developed. The sensor was tested with calcium alginate beads encapsulating iron ions and ascorbic acid as active compounds. The prediction and measurement potential of this sensor was improved by developing a thermodynamic model that allows obtaining kinetic parameters that will allow suitable encapsulation design for subsequent release.

## 1. Introduction

Even though in recent years the general and scientific interest in nutrition, digestion, and what role they play in our body has increased, there is still much work to be carried out in the field of developing sensors and techniques that are capable of identifying and quantifying the chemical species involved in these processes. Until now, researchers have been monitoring the release of microencapsulated or nanoencapsulated compounds in liquid media in a static process, taking a sample of the medium and measuring it by a spectrophotometer [1,2,3,4,5], proximate analysis (Folin–Ciocalteu method) [6,7,8,9,10], or HPLC [11], which are static and invasive measurements.

According to WHO (Geneva, Switzerland), at present, more than 25% of the world’s population suffers from anaemia, mainly due to iron deficiency [12,13]. Therefore, it can be considered that iron deficiency is the most common and widespread nutritional disorder; it is a deficiency disease that mainly affects the health of children and women, mainly in developing countries, but also in industrialized countries [13]. Iron from the diet may be available as heme (organic iron), or as non-heme (inorganic iron). Heme iron is found mainly in meats (myoglobin) and blood (hemoglobin) and, on the other hand, the main sources of non-heme iron are of plant origin, milk and eggs, and it is found mostly in its oxidized form (Fe^+3^) and bound to various macromolecules [14]. Although non-heme iron is the most predominant form in the usual diet (80–90% of total iron), it is the one with the lowest bioavailability since its absorption may be interfered with by other factors in the diet, such as phytates, calcium, or mucin [15]. On the other hand, heme iron only represents 10–20% of the iron present in the diet, but its absorption is more efficient [16]. The absorption of non-heme iron requires its reduction from the ferric state (Fe^+3^) to ferrous (Fe^+2^) that begins in the gastric acid environment, because iron in the ferric state is not absorbable [14]. There are chemical species with reducing capacity (antioxidants) that also can reduce iron, such as ascorbic acid (vitamin C), cysteine and histidine.

For all of the above, reinforcing the diet with foods that include ferrous salts, or ferric salts accompanied by species with antioxidant activity, which can be dosed throughout the digestive system, could increase the assimilation of iron and thus reduce the problems associated with diets with little or no heme iron. In this context, it is important to study the encapsulation of ferrous salts, or ferric salts accompanied by species with antioxidant activity, and their release under gastrointestinal conditions. This encapsulation has several utilities; firstly, it ensures the release throughout the entire small intestine, mainly in the duodenum and jejunum. In addition, it prevents the reaction of ferric and ferrous salts with other chemical species of gastric juices, masking the metallic taste of these salts and eliminating the staining effect that some vitamin complexes have on food and teeth [17].

Encapsulation is a process in which certain active compounds are included within a protective matrix in order to control their release and/or protect them from degradation processes such as oxidation, evaporation or thermal degradation [18,19,20]. In the search for suitable matrices, both calcium alginate hydrogels (alginate-Ca (II)) and powders obtained by spray drying have been widely studied [21,22,23]. Alginate is an anionic polyeletrolite, made up of β-D-manuronic (M) and α-L-guluronic (G) residues linked by α bonds (1 → 4), and generate a hydrogel matrix, due to the ionic crosslinking of the G blocks with di- or trivalent cations such as Ca^2+^ or Ce^3+^ (among others), forming the structure known as “egg box” [24]. Ca (II)-alginate is non-toxic, biodegradable and biocompatible, the process for the production of beads being cheap, easy and versatile, in addition to offering extremely manageable operating conditions. However, it has certain disadvantages such as the loss of encapsulated compounds, low mechanical resistance, large pore size [25] and high-water activity, which makes it susceptible to microbial deterioration [26]. This is why the coupled application of a conservation treatment such as hot air drying will improve the stability of the system for the subsequent release of bioactive compounds.

To describe and estimate a mass transport process, such as the release of encapsulated compounds from the bead to the medium, irreversible thermodynamics can be used. They have been applied successfully in a salting cheese [27], orange peel and potato dried with hot air drying combined with microwaves [28,29], potato and pork freezing [30,31], glass transition and driving forces in the water nucleation process [32,33], estimation of crystal internal energy in meat [34] and a state diagram to determine the quality of foodstuff [35].

There are different spectrophotometric techniques widely used for the control, characterization and monitor processes in the food industry [19,36,37]. Among the most promising techniques is radiofrequency (RF) spectroscopy. It is important to highlight that any physical, chemical, and compositional changes that occur in the hydrogel matrices will generate changes in the dielectric properties in this section of the spectrum, which is why this technique represents a great advance for monitoring the release of active substances during the digestive process. The application of this technique to the development of sensors has been shown to be useful and reliable with sensors capable of performing quality controls such as detecting the deep pectoral myopathy [38] and white striping [39] in poultry, estimating the meat batter composition [40], determining the pork [41] and chicken [42,43] meat quality, assessing the protein denaturation in beef [44], identifying the variety of an apple [45], segregating the different tissues of a mandarin [46] and monitoring the freezing process of potatoes [30].

The aim of this paper is to develop a sensor for measuring the release of encapsulated active compounds, in different media, based on dielectric properties measurement in the radiofrequency range.

## 2. Materials and Methods

### 2.1. Buffer

The buffer solution, characterized by a pH of 3.8 and a 0.05 M concentration, was prepared from acetic acid and sodium acetate (Scharlab S.L., Barcelona, Spain). The purpose of the reactants was to lower the alginate pKa, getting it to be negatively charged, ensuring an optimal interaction between alginate and calcium [25].

### 2.2. Reagents

The reagents used in the preparation of the samples, together with the buffer, were calcium chloride (CaCl_2_) (Scharlab SL, Barcelona, Spain), sodium alginate (Panreac Química SLU, Castellar del Valles, Barcelona, Spain) with a purity of 90%, iron-protein- succinylate (Ferplex, Italfarmaco SA, Madrid, Spain), with a purity of 40 mg/15 mL, L(+)-Ascorbic acid (Scharlab SL, Barcelona, Spain) and hydrochloric acid (1N) (Panreac Química SLU, Castellar del Valles, Barcelona, Spain).

### 2.3. Solutions of Different pH

For the preparation of the solutions at different pH 3 and 4.7, hydrochloric acid was added to double-distilled water until the final pH was reached. pH was monitored by pH meter (S20 SevenEasy TM, Mettler Toledo, Greifensee, Switzerland).

### 2.4. D Printing Material: Acrylonitrile Butadiene Styrene (ABS)

This material (FrontierFila, Shenzen, China) was printed with the following parameters: 235 °C extrusion temperature, 90 °C bed temperature, 100% filling, 40 mm/s print speed and a layer height of 0.1 mm. The filament has a diameter of 1.75 mm according to the specifications of the printer extruder.

### 2.5. Experimental Procedure

A specific measurement system for dielectric properties was designed to allow continuous measurements during the release of active compounds. This measuring system consists of an outer shell and a measuring tank, both printed by 3D printing. Subsequently, the parts and metals chosen to obtain the dielectric properties sensor were assembled and connected to an impedance analyzer in order to perform measurements.

Standard solutions were prepared with known amounts of iron-protein-succinylate and L(+)-Ascorbic acid. In both cases, they were prepared at both pHs (3 and 4.7). Standard solutions were prepared with mass fractions from 100 to 500 ppm of iron ion, and of ascorbic acid in mass fractions from 50 to 3000 ppm. Once the standard solutions were prepared, they were measured to analyze the possibility of determining specific amounts of these compounds in different media using the developed sensor.

Standard solutions of active compounds were made from mass fractions of 100–500 ppm of iron ion, and of ascorbic acid in mass fractions from 50–3000 ppm at two pH levels: 3 and 4.7.

After this process of tuning and testing the sensor, three types of beads were made: calcium alginate (alginate beads), calcium alginate with iron-protein-succinylate (iron ion beads), and finally, with L(+)-Ascorbic acid (ascorbic acids beads), all submitted to a drying process at 40 °C and 0.8 bar for 24 h in a vacuum drying oven (Vaciotem-T, Grupo Selecta, Abrera, Barcelona, Spain). Amounts of 0.05 g of beads and 200 µL of medium (solutions at different pH 3 and 4.7) were put into the measuring tank and measurements of dielectric properties were made for two hours. These measurements were taken at 5, 15, 30, 60, 120, 150, 200, 240, 300, 360, 420, 480, 540, 600, 1200, 1500, 1800, 2500, 3600, 4500 s.

Finally, a measure of expansion was made within different media (solutions at both pH 3 and 4.7) of the alginate beads, iron ion beads and ascorbic acid beads.

### 2.6. D printing Protocol

The protocol followed for the design and obtaining of the external shell and the measuring tank is divided into three steps. First, the piece was designed using a 3D design program (Tinkercad, Autodesk, Inc., Mill Valley, CA, USA), in which all the dimensions of the desired pieces were specified (Figure 1). Once the design prototype is obtained, it is sent to the 3D printer (Anet A8), setting all the previously established parameters. For this purpose, the Repetier–Host software was used to control and calibrate the printer, as well as to transmit data in GCode file to be replicated by the 3D printer, previously heated to the optimum temperature established for each material. To convert the 3D digital design into the instructions and steps necessary to achieve the physical design, the Slic3c tool was used to cut the model into layers, generate the paths to fill it and calculate the amount of material that will need to be extruded.

### 2.7. Calcium Alginate Beads Encapsulating Iron-Protein-Succinylate and Ascorbic Acid Preparation Protocol

Alginate and iron ion or ascorbic acids beads were prepared by ionic gelation. For this, it was necessary to prepare a 1:100 solution of sodium alginate (together with the reagent to be encapsulated: iron-protein-succinylate or L(+)-ascorbic acid) and another 1:100 solution of calcium chloride, both prepared with the sodium acetate buffer. A frequency inverter (Inverter DV-700 Panasonic, Osaka, Japan) was installed to control the revolutions per minute of the peristaltic pump (Damova S.L, Barcelona, Spain, model CPM-045B). The peristaltic pump drips the solution formed by sodium alginate and the reagent that was to be encapsulated on the CaCl_2_ solution with a ratio of 1:10, the CaCl_2_ solution was in continuous agitation (IKA^®^ MS3 basic, Staufen im Breisgau, Germany). The speed used to make a correct drip must be based on the percentage of power that the frequency inverter gives to the pump, in this case, it was 30% and the distance between the tip of the needle (Thermo Fisher Scientific Oy, Vantaa, Finland) and the solution of CaCl_2_ is 10 cm.

Once the beads were obtained, they were left under stirring in the CaCl_2_ solution for 15 min to ensure optimum gelation [47]. After 15 min, the beads were extracted from the solution and washed 5 times with distilled water to remove the excess of CaCl_2_.

Once the wet beads of calcium alginate and iron-protein-succinylate or calcium alginate and L(+)-ascorbic acid were formed, they were put into previously weighed crucibles and the total mass was registered. The crucibles were then placed in a vacuum drying oven (Vaciotem-T, JPSELECTA, Barcelona, Spain) at 40 °C and 0.8 bar for 24 h. After 24 h, the dried samples with a water activity (a_w_) less than 0.35 were weighed and subsequently stored in Aqualab^®^ (Pullman, WA, USA) capsules sealed with Parafilm^®^ (Sigma-Aldrich, San Louis, USA) to avoid possible rehydration.

### 2.8. Protocol for the Determination of the Expansion Capacity of the Beads in Different Media

Once the dry beads were obtained, the protocol for determining the capacity of expansion was performed in triplicate: expansion capacity of the control beads in the different media (pH 3 and 4.7), expansion capacity of ascorbic acid beads in the different media (pH 3 and 4.7) and, the expansion of iron ion beads in the different media (pH 3 and 4.7). The beads were subjected to a rehydration process to quantify and replicate the increase in size that they experience inside the sensor when they come into contact with a solution of a specific pH. The assembly consists of a microscope (Juision USB Microscope) connected to a computer (MacBook Air, Apple, Cupertino, CA, USA), using the “Photo Booth” software for taking photos. As a reference distance, a micrometered glass located at the base of the bead was used (Figure 2).

A single bead was placed inside a glass crucible, then 100 µL of the corresponding pH (3 and 4.7) solution was added. The first photo was taken once the liquid phase comes into contact with the bead. After this the first, more photographs were taken at 5, 15, 30, 60, 120, 150, 200, 240, 300, 360, 420, 480, 540, 600 s from that time a measurement was taken every 5 min until 30 min.

The images were analyzed in Photoshop^®^ (CS5, ver. 12, Adobe Systems InCorp., San Jose, CA, USA) by analyzing with the measuring tools the circumference of the 2D image of the bead with a sphere shape and a square millimeter provided by the micrometered glass, in order to transform the measurement from square pixels to square millimeters, thus obtaining the radius of the remainder of the bead sphere, and finally, the volume of each time point.

### 2.9. Determination of Liberation Kinetics

In order to obtain the release kinetics of bioactive compounds in different media, a system for measuring dielectric properties was designed and constructed (See results section). The measurement system was connected to an Agilent 4294A Impedance Analyzer, (Agilent Technologies, Santa Clara, CA, USA) (Figure 3). An open and short calibration was performed. The measurement range was from 40 Hz to 1 MHz. A triplicate of each release process was performed: release of alginate beads in different media (pH 3 and 4.7), release of iron ion beads in different media (pH 3 and 4.7), and finally, release of ascorbic acid beads in different media (pH 3 and 4.7).

## 3. Results

Throughout the last decade, the scientific community has tried to simulate human digestive processes, which allow the design of more efficient foods and drugs, where the design of sensors gains relevance, representing a technological and scientific challenge for the research community. In this context, the design of encapsulates that allow the release of active compounds to fix digestive problems, nutritional deficiencies or diseases, requires the design of digestive simulation sensors to quantify said release. In this context, a sensor was developed for measuring the release of encapsulated active compounds in different media, based on dielectric properties measurement in the radio frequency range.

The sensor consisted of two parts (see Figure 4), an outer shell, and a measuring tank. Figure 4a–c show the measuring tank in plan, elevation and cross-section view, and Figure 4d–f, the outer shell where the measuring tank is fixed to the circuit (shown in plan and cross-section view). The material selected for the final design was ABS, due to its ability to resist acidic and basic pH medium.

The measuring tank was designed with the aim of introducing the two parallel tantalum plates (0.75 cm × 1.5 cm) inside the tank, glued to the inner walls, between which the liquid phase and the beads were located (see Figure 5). In addition, two clamping rectangles were added to the base, which were put into the outer shell to improve its fixation. The measuring tank is connected to the impedance analyzer, as Figure 5 shows, obtaining the complex impedance, being able to transform to complex permittivity, dielectric constant (ε’) and loss factor (ε”), by means of the equations shown in this figure, thanks to the parallel arrangement of the tantalum plates.

Figure 6a and Figure 7a show the swelling of alginate beads with iron ion and ascorbic acid, respectively, when they are put into both pH media. The swelling of both types of beads in the different media causes a subatmospheric pressure variation inside the beads that causes the entry of media from the outside. The liquid phase (*LP*) flux entering the beads is calculated from the volume variation, using the following equation:(1)$$J_{LP}=\frac{∆V\cdot \rho_{LP}}{S\cdot t}$$
where ∆*V* is the volume change, *ρ_LP_* is the density of *LP* (being considered equal to the density of water since the content of solutes is very low), *S* is the bead surface and *t* is the time in seconds. The evolution of the LP flux entering the alginate beads with iron ion and ascorbic acid is shown in Figure 6b and Figure 7b.

The beads immersed in an aqueous medium will release the active compound, with high ionic strength (iron ion) or moderate (ascorbic acid), varying the dielectric properties of the medium depending on the concentration of these chemical species. It will be necessary to determine the relationship of this compound with the dielectric properties and the swelling of the beads, and thus quantify the release of the active compound from the encapsulation. In Figure 6a and Figure 7a, it is possible to observe how the swelling of the beads relaxes after 360 s, the volume variation remains constant approximately over 60% in the case of iron ion at pH 3, 70% at pH 4.7, and in the case of ascorbic acid, 80% in both cases.

In order to obtain the calibration of the release measurement system, standard solutions of active compounds were made from mass fractions of 100–500 ppm of iron ion, and of ascorbic acid in mass fractions from 50–3000 ppm at two pH levels: 3 and 4.7. Considering the nature of the chemical species released, with a high or moderate ionic strength, a relation between the content of these species and the dielectric properties should be observed, in the section of the electromagnetic spectrum comprised in the alpha dispersion, the counterion effect. For this reason, the spectra in the region of 40 Hz to 1 kHz were analyzed, observing a greater relationship at 200 Hz, mainly in the reactance. Moreover, it was observed that the pH had no significant effect on the measurements of dielectric properties at frequencies of kHz. For this reason, all the measurements were grouped in the same graph that related, on the one hand, the mass fraction of iron ion with regard to the reactance, and on the other hand, the amount of ascorbic acid with the reactance (Figure 8). This figure shows that there is a linear relationship between the mass fraction of iron and the reactance at 200 Hz. Ascorbic acid also had a linear relationship with the reactance at 200 Hz.

During the releasing process of each active compound in the measuring tank, the evolution of the reactance was obtained at 200 Hz, for each condition of the external liquid phase, as was specified in the materials and methods section. These measurements are shown in Figure 9a.

With the values of reactance measured at 200 Hz, and from the calibration of standard solutions shown in Figure 8, the dielectric measurement was transformed into the concentration of each chemical species in the liquid phase. As can be seen in Figure 9b, the main quantity of iron ion and ascorbic acid is released in the first 450 s, reaching the asymptote of release at approximately 900 s for the two active compounds.

In order to obtain the release flux of each active compound, it is necessary to establish mass balances to the system bead/liquid phase. Figure 10 shows an outline of the bead/liquid phase system and the mass balances applied to this system. From these balances, it is possible to obtain the variation in mass fraction of each active compound, inside beads, during the active compound release process.

Once the mass fractions of each active compound are obtained, during the release process, it is possible to calculate its flux with the following equation.
(2)$$J_{i}=\frac{∆m_{i}^{B}}{∆t\cdot S^{B}\cdot M^{i}}$$
where *J_i_* es de molar flux (mol_i_/s m^2^), ∆*m_i_^B^* is the active compound mass variation in bead (g), during ∆*t* time (s), *S^B^* (m^2^) is the bead surface at this process time, and *M^i^* is the molecular weight of the active compound (g/mol).

Figure 11 shows the active compound fluxes release from the beads in a liquid media.

The engine that produces the release of active compounds is the chemical potential gradient at the bead–liquid phase interface. Within the chemical potential, there are chemical or mechanical gradients that will affect mass transport. The main engines for the active compound transport are the gradient concentration of the chemical species and the pressure variation induced by the bead swelling, therefore, the gradient of the chemical potential of each active compound may be defined by the Gibbs–Duhem expression as [48]:(3)$$∆\mu_{i}=RT\;ln\frac{c_{i}^{LP}}{c_{i}^{B}}+v_{i}\cdot ∆P$$
where ∆*μ_i_* is the chemical potential of each active compound, *R* is the ideal gas constant (8.314 J/mol K), *T* is the temperature (K), *c_i_* is the molar concentration (mol_i_/m^3^), *ν_i_* is the specific volume of *i* and ∆*P* is the pressure gradient between bead and LP.

The relationship between the molar flux and the chemical potential is defined by the first Onsager reciprocity relation, according to the following equation [49]:(4)$$J_{i}=L_{i}\cdot ∆\mu_{i}$$
where *L_i_* is the phenomenological coefficient expressed in mol^2^/J s m^2^. This phenomenological coefficient describes the ability of a chemical species to transport itself through a medium. From the mass flux and the chemical potential, it is possible to calculate the phenomenological coefficient. However, with the experimental data obtained, it is only possible to calculate the term of concentrations of the chemical potential, as shown in Equation (5).
(5)$$∆\mu_{i}^{*}=RT\;ln\frac{c_{i}^{LP}}{c_{i}^{B}}$$
where $∆\mu_{i}^{*}$ is the chemical potential of each active compound, considering only the concentration term.

In Figure 12, the chemical potential is shown without the mechanical term; however, it is possible to assume that when the swelling of the capsule is negligible, the mechanical term will be as well.

Considering the mechanical term negligible when the bead swelling is negligible, it is possible to calculate the phenomenological coefficient for each type of bead. The result from this is 5.6 ± 0.7 × 10^−10^ mol^2^/J s m^2^ for iron ion at pH 3, 2.0 ± 0.3 × 10^−10^ mol^2^/J s m^2^ for iron ion at pH 4.7, 3.7 ± 0.3 × 10^−10^ mol^2^/J s m^2^ for ascorbic acid at pH 3 and 9.8 ± 0.8 × 10^−10^ mol^2^/J s m^2^ for ascorbic acid at pH 4.7.

Using these phenomenological coefficients, it is possible to obtain the gradients of the chemical potential during the entire release process, and with it and using Equation (3), the mechanical term can be obtained.

Figure 13a,b shows the evolution of the mechanical term throughout the active compound release process. This mechanical term is induced by the swelling of the beads. This phenomenon occurs since, in their preparation, the beads undergo a dehydration process, which generates a drastic shrinkage and vitrification that causes storage of mechanical energy, which can only be released when hydrated again, changing to a rubbery state, and recovering its native elasticity.

The mechanical term causes a flux of external liquid phase to enter, and slows the transport of active compound to the outside of beads.

This phenomenon is observed in Figure 13c,d, where the mechanical term is compared with bead swelling. It is possible to observe that both evolve together and stop at the same time, the moment where they reach the maximum swelling. Moreover, these figures show that the mechanical terms are higher in the iron beads, although the ascorbic acid beads have more swelling, this may be due to the fact that the iron ions have a greater ionic strength, which will affect the formation of the beads and which are based on ionic gelation.

## 4. Conclusions

A system for measuring the release of microencapsulated active compounds was developed from impedance measurements in the radio frequency range. This system was tested with calcium alginate beads encapsulating iron ions and ascorbic acid as active compounds.

The prediction and measurement potential of this sensor was improved by developing a thermodynamic model that allows quantifying kinetic design parameters such as the phenomenological coefficient.

The sensor was tested in an aqueous liquid medium in the pH range in which digestive media are found in the stomach phase, in order to determine interferences in impedance measurements in the radio frequency range, showing great precision in the measurements and no interference with the medium. However, an effect of pH was observed on the swelling processes of the beads, possibly induced by ion–ion relationships in the gel matrix of calcium alginate.

The phenomenological coefficients obtained are in the same range of values, for iron (2–5.6 × 10^−10^ mol^2^/J s m^2^) and ascorbic acid (3.7–9.8 × 10^−10^ mol^2^/J s m^2^), showing adequate encapsulation design, since it will release a similar proportion of iron and ascorbic acid, which will act as an antioxidant, maintaining the reduced state of iron and, therefore, facilitating its absorption.

## Figures and Tables

**Figure 1 sensors-21-05781-f001:**
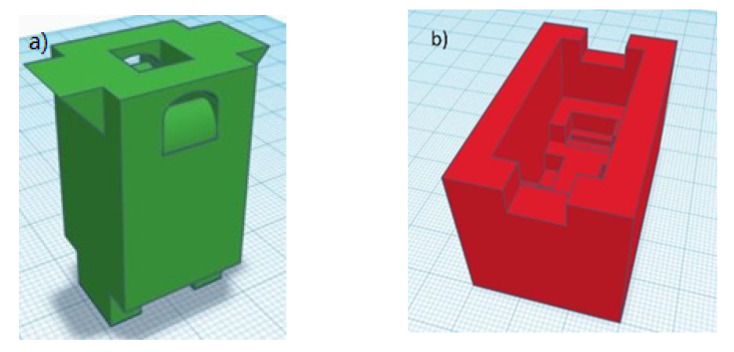
Design of the sensor: (**a**) measuring tank (**b**) outer shell. The ground shows millimeter graph paper.

**Figure 2 sensors-21-05781-f002:**
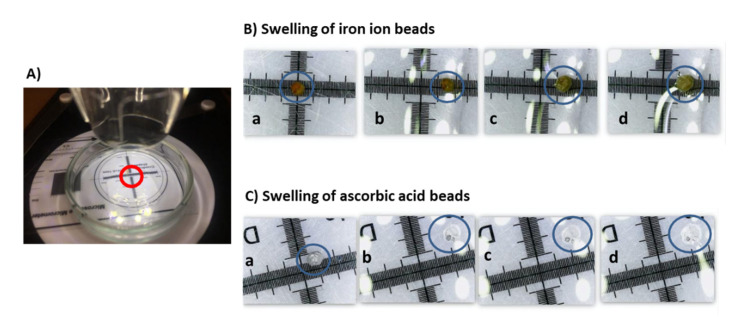
(**A**) Crucible containing the bead and the liquid phase to analyze the swelling by a microscope; (**B**) Swelling of an iron ion bead at pH 4.7 at the following times: (a) 0 min; (b) 0.367 min; (c) 6 min; (d) 30 min; (**C**) Swelling of an ascorbic acid bead at pH 4.7 at the following times: (a) 0 min; (b) 0.6 min; (c) 4 min; (d) 30 min.

**Figure 3 sensors-21-05781-f003:**
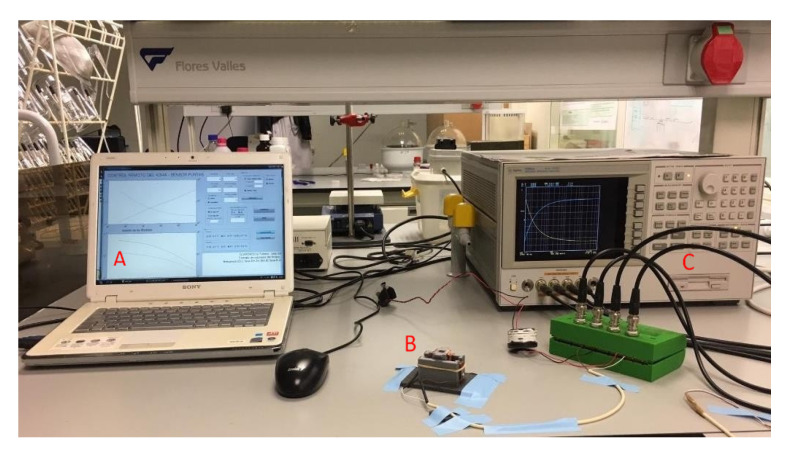
Set-up used to determine release kinetics. (A) Computer that collects the data from the impedance analyzer; (B) measuring tank and outer shell; (C) Impedance analyzer.

**Figure 4 sensors-21-05781-f004:**
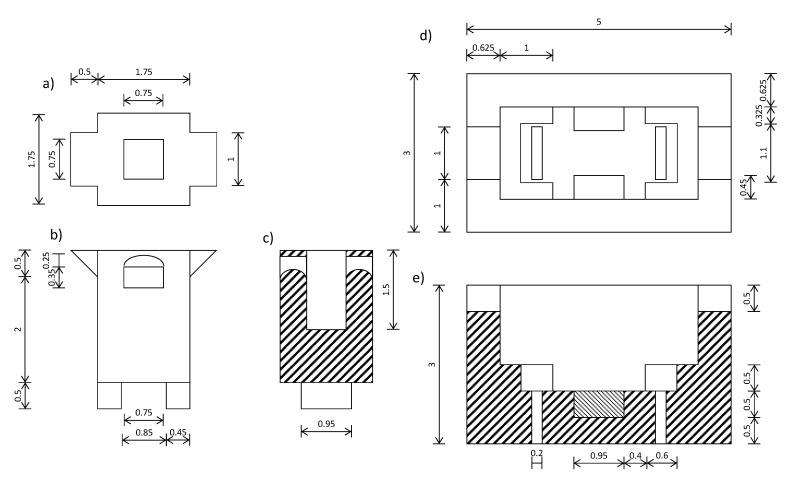
(**a**) Plan view of the measuring tank; (**b**) Elevation view of the measuring tank; (**c**) Cross-section section of the measuring tank; (**d**) Plan view of the outer shell; (**e**) Cross-section of the measuring tank. The dimensions of the whole figure are in centimeters.

**Figure 5 sensors-21-05781-f005:**
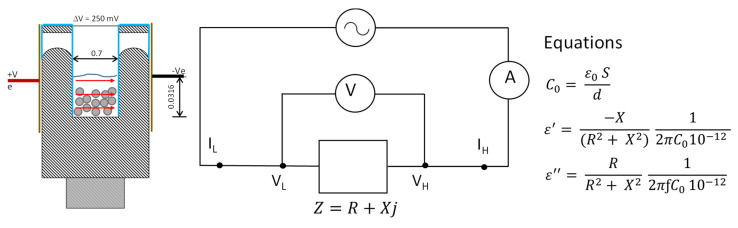
Detail of sensor of Impedance measurement in radiofrequency range, to determine the release of encapsulated active compound, the electric circuit of sensor and the equations to determine the permittivity.

**Figure 6 sensors-21-05781-f006:**
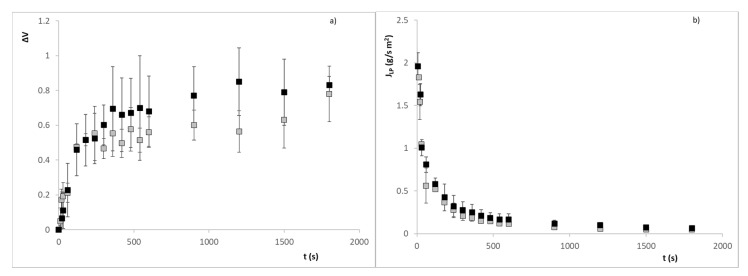
(**a**) Evolution of the volume variation of alginate beads with iron ion; (**b**) Evolution of the LP flux entering the calcium alginate beads with iron ion. Symbol ■ refers to pH = 3 and ■ refers to pH = 4.7.

**Figure 7 sensors-21-05781-f007:**
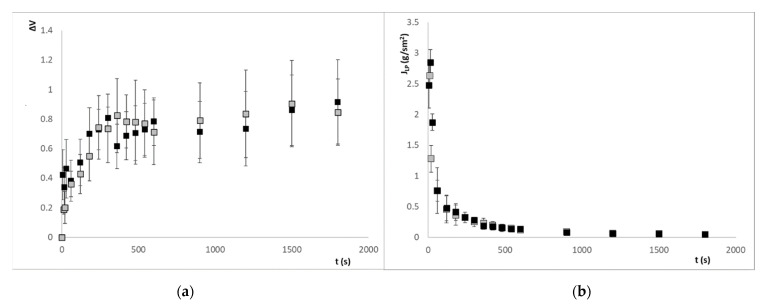
(**a**) Evolution of the volume variation of alginate beads with ascorbic acid; (**b**) Evolution of the liquid phase flux entering the calcium alginate beads with vitamin C. Symbol ■ refers to pH = 3 and ■ refers to pH = 4.7.

**Figure 8 sensors-21-05781-f008:**
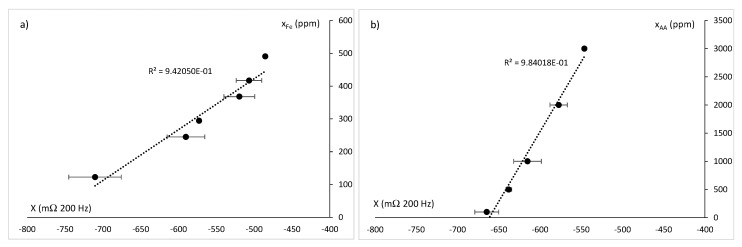
(**a**) Mass fraction of iron ion (ppm) vs. reactance at 200 Hz (mΩ); (**b**) Mass fraction of ascorbic acid (ppm) vs. reactance at 200 Hz (mΩ). Notation is according to IuPAC.

**Figure 9 sensors-21-05781-f009:**
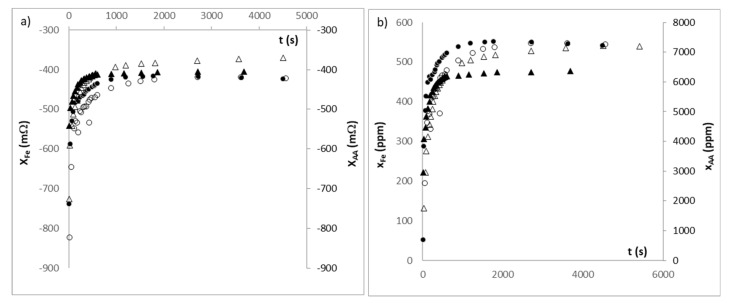
(**a**) Evolution of reactance at 200 Hz (mΩ) throughout the release process; (**b**) Evolution of mass fraction of each active compound (ppm) throughout the release process. Where (●) is iron ion released at pH of 3; (○) is iron ion released at pH of 4.7; (▲) is ascorbic acid released at pH of 3 and (∆) is ascorbic acid released at pH of 4.7.

**Figure 10 sensors-21-05781-f010:**
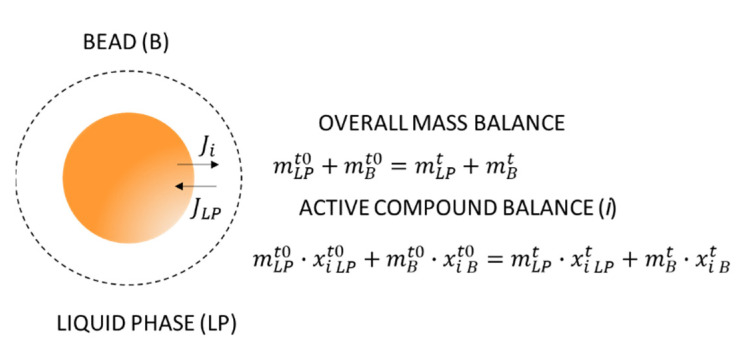
Outline of the bead in the liquid phase during the release of the active compound, with details of the fluxes and mass balances applied to the bead–liquid phase system. Where the subscripts: liquid phase (LP), bead phase (B) and active compound (i), and superscripts: t0 (initial time) and t (process time).

**Figure 11 sensors-21-05781-f011:**
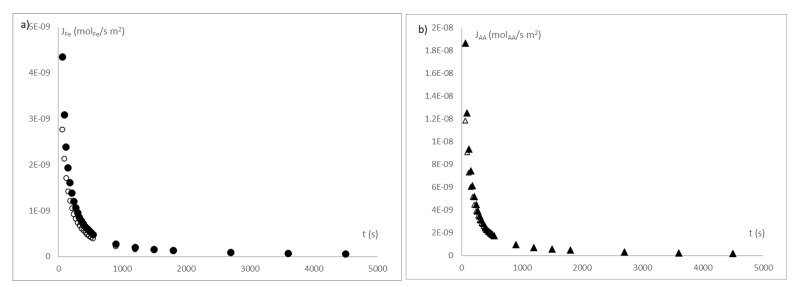
(**a**) Evolution of iron release flux (J_Fe_)and (**b**) Evolution of ascorbic acid release flux (J_AA_). Where (●) is iron ion released at pH of 3; (○) is iron ion released at pH of 4.7; (▲) is ascorbic acid released at pH of 3 and (∆) is ascorbic acid released at pH of 4.7. Notation is according to IuPAC.

**Figure 12 sensors-21-05781-f012:**
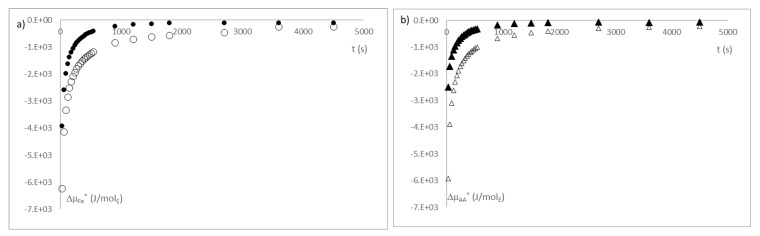
(**a**) Gradient of partial iron chemical potential and (**b**) gradient of partial ascorbic acid chemical potential in the bead–liquid phase interface. Where (●) is iron ion released at pH of 3; (○) is iron ion released at pH of 4.7; (▲) is ascorbic acid released at pH of 3 and (∆) is ascorbic acid released at pH of 4.7. Notation is according to IuPAC.

**Figure 13 sensors-21-05781-f013:**
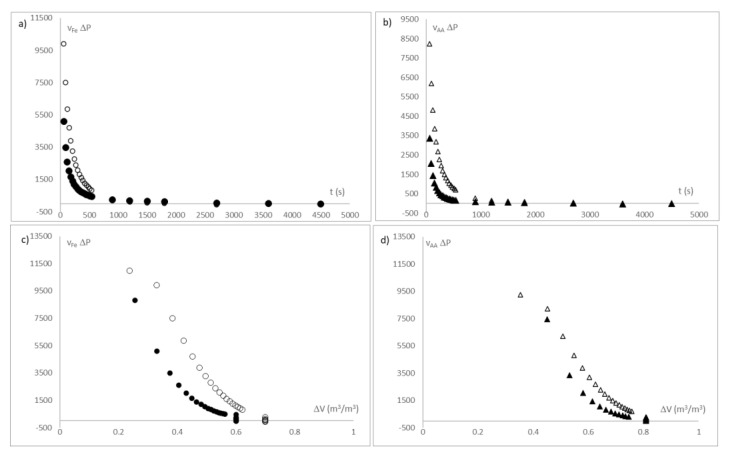
(**a**,**b**) represents the evolution of the mechanical gradient of (**a**) the iron ion beads and (**b**) the ascorbic acid beads at 3 and 4.7 pH; (**b**,**c**) is the relationship between the mechanical gradient and volume variation of (**c**) the iron ion beads and (**d**) the ascorbic acid beads at 3 and 4.7 pH. Where (●) is iron ion released at pH of 3; (○) is iron ion released at pH of 4.7; (▲) is ascorbic acid released at pH of 3 and (∆) is ascorbic acid released at pH of 4.7.

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
