# Peer review of "New Sensor to Measure the Microencapsulated Active Compounds Released in an Aqueous Liquid Media Based in Dielectric Properties in Radiofrequency Range"

_sensors, 2021, doi:10.3390/s21175781_

Round 1

Reviewer 1 Report

The authors of the paper : New sensor to measure the microencapsulated active compounds released in an aqueous liquid media based in dielectric
properties in radiofrequency range present some interesting results in a field of great importance. Few minor corrections can be considered in order to improve the quality of the manuscript: 

L14: Iron deficiency or iron deficiency ? what did you mean ?

L65: ref. 18 appear before 17, please check and re-arrange 

L125: please provide a reference 

L138: standard with S 

L169: Fig.1 provide a scale in mm or cm 

L193: explain: aw 

Fig.2 try to improve the quality of the detail images and to give a general explanation of the figure 2 

L224: what was the purpose of the calibration ? 

L254: General explanation of figure 4 ...and other from the paper 

L386: provide a reference for first Onsager relation 

re-structure the conclusions with experimental results 

Author Response

The authors of the paper : New sensor to measure the microencapsulated active compounds released in an aqueous liquid media based in dielectric
properties in radiofrequency range present some interesting results in a field of great importance. Few minor corrections can be considered in order to improve the quality of the manuscript: 

Thank you for your review and your comments

L14: Iron deficiency or iron deficiency ? what did you mean ?

Sorry, it was a mistake, it has been fixed.

L65: ref. 18 appear before 17, please check and re-arrange 

Sorry, it was a mistake, it has been fixed.

L125: please provide a reference 

Sorry, but the printer the extruder specifications for the ABS printing material from FrontierFila company, was not published, it was in the packing box.

L138: standard with S 

It has been changed

L169: Fig.1 provide a scale in mm or cm 

It has been included next sentence “The ground shows millimeter graph paper”.

L193: explain: aw 

aw represents water activity and we have included in the text “…with a water activity (aw)”

Fig.2 try to improve the quality of the detail images and to give a general explanation of the figure 2 

Thanks for the comment, we have improved the explanation L217-222:

“…The images were analyzed in Photoshop® (CS5, ver. 12, Adobe Systems InCorp. USA) by analyzing with the measuring tools the circumference of the 2D image of the bead with a sphere-shape and a square millimeter provided by the micrometered glass, in order to transform the measurement from square pixels to square millimeters, thus obtaining the radius of the remainder of the bead sphere, and finally, the volume of each timepoint.”

L224: what was the purpose of the calibration ? 

The impedance analyzer requires that it be measured under conditions of insulation (air) and total electrical conductivity (short), to eliminate the effect of resistivity and resistance of the equipment circuit and the cables that connect the equipment to the sensor.

L254: General explanation of figure 4 ...and other from the paper 

Thanks for the comment, we have improved the explanation L247-250:

“The sensor consisted of two parts (see figure 4), an outer shell, and a measuring tank. Figure 4 a, b and c shows the measuring tank in plan, elevation and cross-section view, and figure 4 d, e and f, the outer shell where the measuring tank is fixed to the circuit (shown in plan and cross-section view). The material selected for the final design was ABS, due to its ability to resist acidic and basic pH medium.”

L386: provide a reference for first Onsager relation 

We have included next reference: “Gambar, K. and Markus, F. (1993) “On  the  Global  Symmetry  off Thermodynamics  and  Onsager's  Reciprocity” J. Non-Equilib. Thermodyn. 18, pp 51-57.

re-structure the conclusions with experimental results 

Thanks for the comment, we have included some experimental results in the conclusions, specifically the phenomenological coefficient

Reviewer 2 Report

The submitted manuscript is interesting and considers important issues for sensing and health monitoring. Unfortunately, I have found a few problems that have to be explained before publishing.

  1. You should give more details about the setup prestented at Fig. 3. You claim a use of radiofrequencies but the presented results are limited mainly to acoustic frequencies. There is limited information about measurement procedure and amplitude of excitation signal.
  2. Figure 5 presents permitivities epsylon' and epsylon '' but there is no clarification in the text. You have to improve it.
  3. Figures 6, 7, 8 and 9, 12, 13 have to small numbers and you use "," instead of "." to separate integer number from the fractions.
  4. Your theoretical consideration should be consise. t means that you have to correlate dielectric constant with the molar flux. It is not clear in the present form.
  5. You should give discussion in the conclusions if you can develop the presented method to monitor the presented process in human body.

Author Response

Reviewer 2:

Comments and Suggestions for Authors

The submitted manuscript is interesting and considers important issues for sensing and health monitoring. Unfortunately, I have found a few problems that have to be explained before publishing.

Thank you for your review and your comments

  1. You should give more details about the setup prestented at Fig. 3. You claim a use of radiofrequencies but the presented results are limited mainly to acoustic frequencies. There is limited information about measurement procedure and amplitude of excitation signal.

Thanks for the comment, in figure 3 an impedance spectrometer is shown, to analyze the effect of a photon flux crossing a liquid dielectric material. Therefore, the electromagnetic spectrum is analyzed, specificly in the 40 Hz to 1 MHz frequency range. In electromagnetic spectrum this range is inside the radiofrequency dispersions range (counterion and maxwell-wagner). The electromagnetic field is made by a difference tension of 250 mV.

  1. Figure 5 presents permitivities epsylon' and epsylon '' but there is no clarification in the text. You have to improve it.

Thanks for the comment, in figure 5 the transformation of complex impedance to complex permittivity, dielectric constant (e’) and loss factor (e’’), is shown, and this explanation has been improved in the text.

  1. Figures 6, 7, 8 and 9, 12, 13 have to small numbers and you use "," instead of "." to separate integer number from the fractions.

Thanks for the comment, we have changed the figures.

  1. Your theoretical consideration should be consise. t means that you have to correlate dielectric constant with the molar flux. It is not clear in the present form.

The idea of this work was to develop a system for measuring dielectric properties, in this case, reactances, to be able to determine concentrations in ppm of chemical species with a certain ionic strength, and those measurements, previously calibrated and transformed into concentration, relate them to a thermodynamic model that would allow designing encapsulations of active compounds, for that reason, the only related dielectric measure has been the reactance with the measurements of iron and ascorbic acid.

  1. You should give discussion in the conclusions if you can develop the presented method to monitor the presented process in human body.

Thanks for the comment, we have included a paragraph in the conclusions, joining the results with the effect of the work in the human body: “The phenomenological coefficients obtained are in the same range of values, for iron (2-5.6 · 10-10 mol2/J s m2) and ascorbic acid (3.7-9.8 · 10-10 mol2/J s m2), showing adequate encapsulation design, since it will release a similar proportion of iron and ascorbic acid, which will act as an antioxidant, maintaining the reduced state of iron and therefore, fa-cilitating its absorption.”

Round 2

Reviewer 2 Report

The authors have answered my questions. I don't have more remarks.